

# Automatic detection of snow avalanches in continuous seismic data using hidden Markov models

Matthias Heck[1], Conny Hammer[2], Alec van Herwijnen[1], Jürg Schweizer[1], and Donat Fäh[2]

[1]WSL Institute for Snow and Avalanche Research SLF, Davos
[2]Swiss Seismological Service SED, ETH Zurich, Zurich

*Correspondence to:* Matthias Heck (matthias.heck@slf.ch)

**Abstract.**

Snow avalanches generate seismic signals as many other mass movements. Detection of avalanches by seismic monitoring is highly relevant to assess avalanche danger. In contrast to other seismic events, signals generated by avalanches do not have a characteristic first arrival nor is it possible to detect different wave phases. In addition, the moving source character of avalanches increases the intricacy of the signals. Although it is possible to visually detect seismic signals produced by avalanches, reliable automatic detection methods for all types of avalanches do not exist yet. We therefore evaluate whether hidden Markov models (HMMs) are suitable for the automatic detection of avalanches in continuous seismic data. We analyzed data recorded during the winter season 2010 by a seismic array deployed in an avalanche starting zone above Davos, Switzerland. We first visually inspected the data and identified more than 200 events we assume to be generated by avalanches. Since most of the data consists of noise we first applied a simple amplitude threshold to reduce the amount of data. As first classification results were unsatisfying, we analyzed the temporal behaviour of the seismic signals for the whole data set and found that there is a high variability in the seismic signals. We therefore applied further post-processing steps to reduce the number of false alarms by defining a minimal duration for the detected event, implementing a voting based approach and analyzing the coherence of the detected events. We obtained the best classification results for events detected by at least 5 sensors and with a minimal duration of $12\,\mathrm{s}$. These processing steps allowed identifying two known periods of high avalanche activity, suggesting that HMMs are suitable for the automatic detection of avalanches in seismic data. However our results also showed that more sensitive sensors and more appropriate sensor locations are needed to improve the signal-to-noise ratio of the signals and therefore the classification.

## 1 Introduction

During the winter season, snow avalanches may threaten people and infrastructure in mountainous regions throughout the world. Avalanche forecasting services therefore regularly issue avalanche bulletins to inform the public about the avalanche conditions. Such an avalanche forecast requires meteorological data, information about the snowpack and avalanche activity data. The latter are mostly obtained through visual observations requiring good visibility. Avalanche activity data are therefore often lacking during periods of intense snowfall, which are typically the periods when they are most important for forecasting.





A possible alternative approach to determine the avalanche activity is to use a seismic monitoring system (e.g. van Herwijnen and Schweizer, 2011a).

Seismic monitoring systems are well suited to detect mass movements such as rockfalls, pyroclastic flows and snow and ice avalanches (Faillettaz et al., 2015; Podolskiy and Walter, 2016; Caplan-Auerbach and Huggel, 2007; Suriñach et al., 2005; Zobin et al., 2009). The ability to detect snow avalanches through seismic methods was first demonstrated in the 1970s. St. Lawrence and Williams (1976) and Harrison (1976) deployed geophones near avalanche paths and manually identified signals generated by avalanches in the seismogram. They showed that the seismic signature of avalanches differs from other seismic events such as earthquakes or nearby blasts. A more in-depth analysis of seismic signals generated by avalanches was performed 20 years later, identifying typical characteristics in both the time and time-frequency domain (Kishimura and Izumi, 1997; Sabot et al., 1998). Using automatic cameras to film avalanches, Sabot et al. (1998) showed that specific features in the seismic signals were related directly to changes in the flow of the avalanche; these findings were confirmed by Suriñach et al. (2000) and Suriñach et al. (2001). Since then, seismic signal characteristics were used to estimate specific properties of single avalanches such as the flow velocity (Vilajosana et al., 2007a), the total energy of the avalanche (Vilajosana et al., 2007b) or the runout distance (Pérez-Guillén et al., 2016; van Herwijnen et al., 2013).

While many studies focused on using seismic signals to better understand the properties of single avalanches, continuous monitoring of avalanche starting zones to obtain more accurate avalanche activity data is of particular interest for avalanche forecasting (e.g. van Herwijnen et al., 2016). Leprettre et al. (1996) deployed three-component seismic sensors at two different field sites and compared seismic signal characteristics to a database including avalanches, helicopters, thunder rolls and earthquakes. Lacroix et al. (2012) improved the seismic monitoring system used by Leprettre et al. (1996) by deploying a seismic array between two known avalanche paths. The array consisted of six vertical component geophones arranged in a circle around a three component geophone in the centre. Using array techniques, Lacroix et al. (2012) determined the release area and the path of manually identified avalanches and estimated their speed. These studies mainly monitored medium and large avalanches. van Herwijnen and Schweizer (2011a), however, deployed seismic sensors near an avalanche starting zone above Davos in the eastern Swiss Alps to also detect small avalanches. They manually identified several hundred avalanche events in the continuous seismic data during four winter months.

While these studies have highlighted the usefulness of seismic monitoring to obtain more accurate and complete avalanche activity data, using machine learning algorithms to automatically detect snow avalanches has thus far remained relatively unsuccessful. Nevertheless the interest in these techniques has been evident for several decades (Leprettre et al., 1998; Bessason et al., 2007; Rubin et al., 2012). The first attempt to automatically detect avalanches focused on using fuzzy logic rules and credibility factors derived from features of the seismic signal in the time and time-frequency domain (Leprettre et al., 1996, 1998). In a first step, the features of unambiguously identified seismic events were analysed including avalanches, blast and teleseismic events. They then formulated several fuzzy logic rules for each type of event to train a classifier used to identify the type of a new unknown seismic event. While the probability of detection (POD), i.e. the number of detected avalanches divided by the total number of observed avalanches, was high ($\approx 90\%$), one of the main drawbacks of this method is the subjective expert knowledge used to derive the fuzzy logic rules and the need to adapt these rules to each individual field site.





Bessason et al. (2007) deployed seismic sensors in several known avalanche paths along an exposed road in Iceland. They used a nearest-neighbour method to automatically identify avalanche events. The method consists of comparing new events with those in a database. Although a 10-year database was used, the identification performed rather poorly. Seismic signals generated by rockfalls and debris flows were wrongly classified as avalanches and vice versa, resulting in a POD of about 65%.

In an attempt to improve the automatic detection of avalanches, Rubin et al. (2012) used a seismic avalanche catalogue presented by van Herwijnen and Schweizer (2011b) and compared the performance of 12 different machine learning algorithms. The PODs of all classifiers were high (between 84% - 93%). However, the main drawback were the high false alarm rates, much too high for operational tasks.

The methods described above are generally difficult to apply at new sites since they require time to build a training data set and/or expert knowledge to define thresholds and rules. To overcome these drawbacks, we investigate using a hidden Markov model (HMM). A HMM is a statistical pattern recognition tool commonly used for speech recognition (Rabiner, 1989) and was first introduced for the classification of seismic traces by Ohrnberger (2001). The advantage of HMMs compared to other classification algorithms is that the time dependency of the data is explicitly taken into account. First studies using HMMs for the classification of seismic data relied on large training data sets (Ohrnberger, 2001). Using this approach Beyreuther et al. (2012) created an earthquake detector.

More recently, Hammer et al. (2012) developed a new approach which only requires one training event. This approach was applied for a volcano fast response system (Hammer et al., 2012) and the detection of rockfalls, earthquakes and quarry blasts on seismic broad-band stations of the Swiss seismological service (SED) (Hammer et al., 2013; Dammeier et al., 2016). Furthermore, Hammer et al. (2017) also detected snow avalanches using data from a seismic broadband station of the Swiss Seismological Service. During a period with high avalanche activity in February 1999, 43 very large confirmed avalanches were detected over a 5 day period with only 4 presumable false alarms. While these detection rates are very encouraging, the investigated avalanche period was exceptional. Furthermore, due to the location of the broadband station at valley bottom, they could not detect small or medium sized avalanches.

For avalanche forecasting information on smaller avalanches is also required. To resolve this issue, we investigate a method to obtain local (i.e. scale of a small valley) avalanche activity data using a seismic monitoring system (van Herwijnen and Schweizer, 2011a). We therefore implement the approach outlined by Hammer et al. (2017) to detect avalanches in the continuous seismic data obtained with the sensors deployed near avalanche starting zones (van Herwijnen and Schweizer, 2011b). Based on the duration of the seismic signals, the majority of these avalanches were likely rather small (van Herwijnen et al., 2016). We therefore used this avalanche catalogue to train and evaluate the performance of a HMM model and discuss limitations and possible improvements.





## 2 Field site and instrumentation

We analysed data obtained from a seismic array deployed above Davos, Switzerland. The field site is located at 2500 m a.s.l. and is surrounded by several avalanche starting zones. The site is easily accessible during the entire year and is also equipped with various automatic weather stations and automatic cameras observing the adjacent slopes.

The array consisted of seven vertical geophones with an eigenfrequency of 14 Hz. Six of the geophones were inserted in a styrofoam housing and placed within the snow, whereas the seventh geophone was inserted in the ground with a spike. The seismic sensors were deployed at the field site from early December until the snow had melted.

The instrumentation was originally designed to record higher frequency signals in order to detect precursor signals of avalanche release (van Herwijnen and Schweizer, 2011b). A 24-bit data acquisition system (Seismic Instruments) was used to

continuously acquire data from the sensors at a sampling rate of 500 Hz. The data were stored locally on a low power computer and manually retrieved approximately every 10 days. A more detailed description of the field site and the instrumentation can be found in van Herwijnen and Schweizer (2011a) and van Herwijnen and Schweizer (2011b).

## 3 Data

Continuous seismic data were recorded from 12 Jan 2010 to 30 April 2010. These data were previously used by van Herwijnen

and Schweizer (2011b), Rubin et al. (2012) and van Herwijnen et al. (2016). The recorded seismic data contains various types of events (van Herwijnen and Schweizer, 2011a), including aeroplanes, helicopters and of course avalanches (Figure 1).

### 3.1 Pre-processing of the seismic data

While during the 107 day period several hundred avalanches were identified, the vast majority of the data consist of noise or seismic events produced by other sources. To reduce the amount of data to process, we applied a simple threshold based event

detection method. It consisted of dividing the continuous seismic data stream in non-overlapping windows of 1024 samples. For each window, a mean absolute amplitude $A_i$ was determined. When $A_i \geq 5\overline{A}$, with $\overline{A}$ the daily mean amplitude, the data were used. Furthermore, a data section of $\Delta t = 60\,\mathrm{s}$ before and after each event was also included to ensure that the onset and coda of each event were incorporated. The amount of data to process was thus reduced by 80%(Figure 2).

Finally, before we used the data for the detection, we applied a bandpass filter. Previous studies showed that seismic sig-

nals generated by avalanches typically have a frequency below 50 Hz (e.g. Harrison, 1976; Schaerer and Salway, 1980). We therefore applied a 4th order Butterworth bandpass filter to our data between 1 and 50 Hz.

### 3.2 Reference avalanche catalogue

van Herwijnen and Schweizer (2011a) visually analysed the seismic time series and the corresponding spectrogram of one sensor and identified $N = 385$ avalanches between 12 January and 30 April 2010. They thus obtained an avalanche catalogue

consisting of the release time $t_i$ and duration $T_i$ for each avalanche. However, only 33 of these avalanches were confirmed by





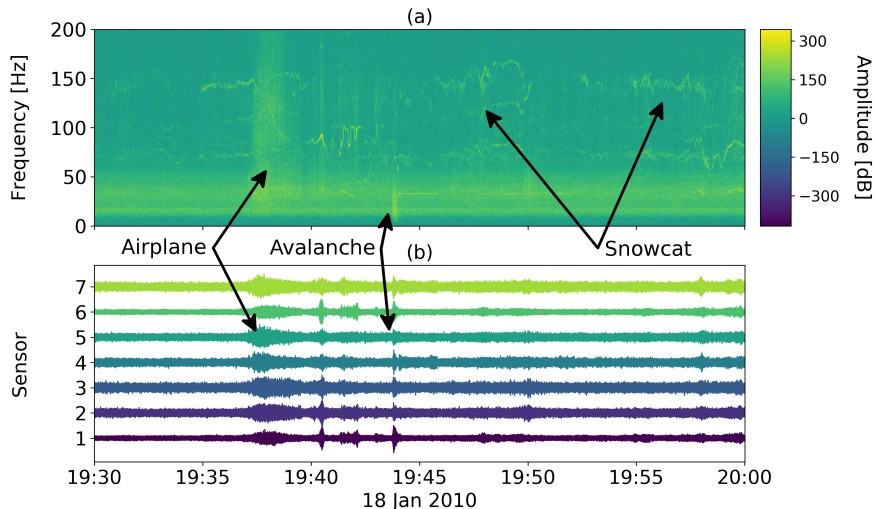

**Figure 1.** (a) Spectrogram of an unfiltered 30 minute time series. (b) Corresponding time series. An airplane as well as an avalanche are visible. Furthermore, noise produced by a snowcat is visible in the second half of the time series.

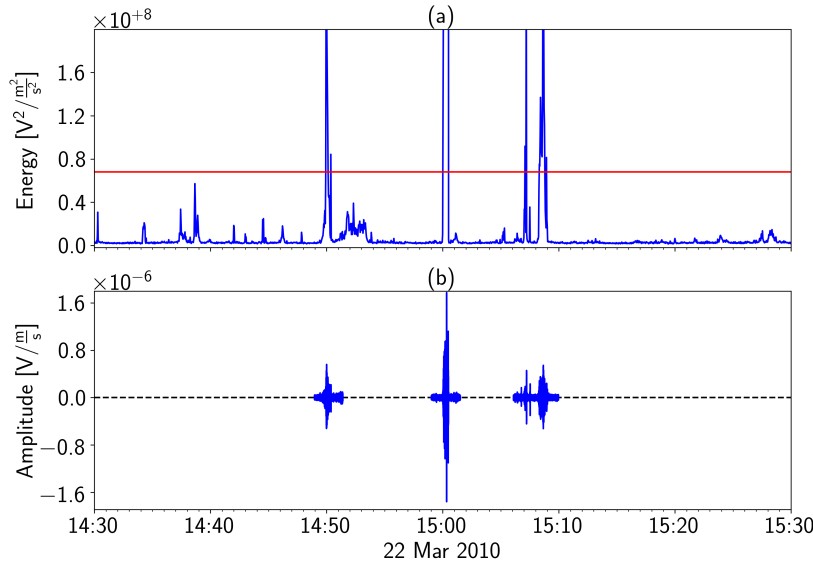

**Figure 2.** Example of the pre-processing. (a) The mean energy values of each window shown as the blue line and the threshold value indicated by the red line. (b) The remaining data cut by the pre-processing step.





**Table 1.** Number of events per probability class $P_{\mathrm{ava}}$ after re-evaluation. The first row shows the possible combinations of subjective probability ($p_e$): $p_e = 1$ it was certain that the event was an avalanche, $p_e = 0.5$ it was uncertain and $p_e = 0$ it was certain that the event was not an avalanche.

| $p_e$ combinations | 0,0,0 | 0.5,0,0 | 1,0,0 | 0.5,0.5,0.5 | 1,1,0 | 1,1,0.5 | 1,1,1 |
|---|---|---|---|---|---|---|---|
| | | | 0.5,0.5,0 | 1,0.5,0 | 1,0.5,0.5 | | |
| $P_{\mathrm{ava}}$ | 0% | 16.6% | 33.3% | 50% | 66.6% | 83.3% | 100% |
| Number of events | 58 | 66 | 48 | 34 | 30 | 27 | 20 |

visual observations, i.e. by field observations or on images obtained from automatic cameras. Hence there remains substantial uncertainty about the nature of the identified events.

To reduce the uncertainty, three of the authors therefore independently re-evaluated this avalanche catalogue. From the 385 avalanches in the original avalanche catalogue only $N_{\mathrm{pre}} = 283$ remained after pre-processing (see section 3.1). By visually 5 inspecting the seismic time series of the seven sensors and the stacked spectrogram for each event, they then assigned a subjective probability $p_e$ to each of these possible avalanche events. Three probabilities were assigned: 1 when it was certain that the observed event was an avalanche, 0 when it was certain that the observed event was not an avalanche and 0.5 when it was uncertain whether the event was an avalanche or not. The probabilities were then combined into seven probability classes depending on the mean probability of each event:

$$P_{\mathrm{ava}} = \frac{1}{3} \sum_{e=1}^{3} p_e \tag{1}$$

with $p_e$ the subjective probability that each assigned to a specific event. In Table 1 the number of events in each probability class is listed. In the reclassified data set, only 20 avalanches merged as certain avalanche by all three evaluators and 58 events were marked as certainly not an avalanche.

The avalanche activity of the season 2010 is shown in Figure 3. Overall, most avalanches were detected in the second half 15 of the investigated period, with two distinct peaks in the avalanche activity around 22 March and 24 April.

## 4 Methods

### 4.1 Hidden Markov model

In this study we used hidden Markov models (HMMs) to detect avalanches in a continuous seismic data set. This statistical classifier models observations (i.e. the seismic time series or its features) by a sequence of multivariate Gaussian probability 20 distribution. The characteristics for the distributions (i.e. mean and covariance) are derived from training sets of known events, so called pre-labelled training sets. Several classes describing different types of events can be implemented in the classifier but each class needs its own training set to determine its unique distribution characteristics. Therefore, the actual classifier consists of several HMMs, one for each class. This classical approach, as used for the classification of seismic time series





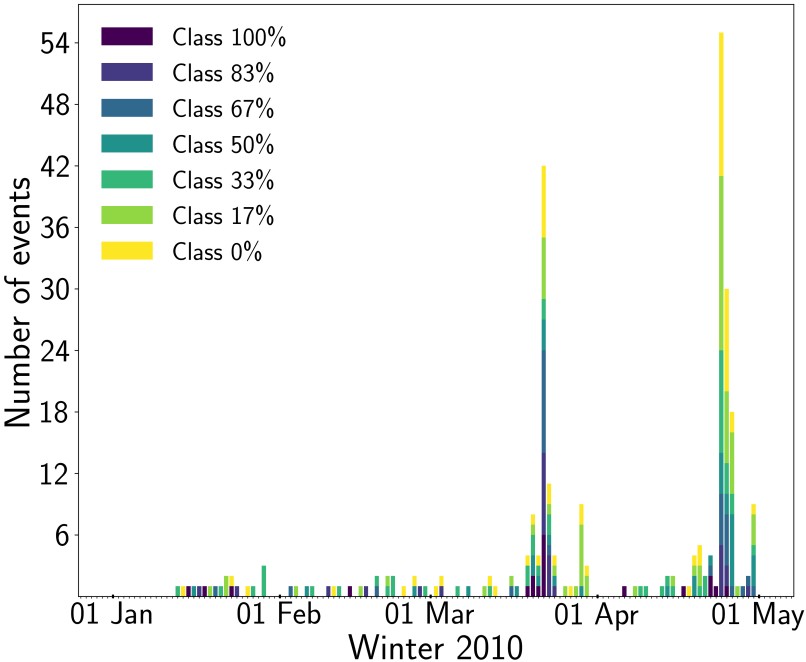

**Figure 3.** Avalanche activity for the winter season of 2010. The different colours indicate the different probability classes derived from the manual detection.

by e.g. Ohrnberger (2001) and Beyreuther et al. (2012), relies on well known pre-labelled training sets. In our case, however, avalanches are rare events and it is nearly impossible or too time consuming to obtain an adequate training set. To circumvent the problem of obtaining sufficiently large training sets, we used a new approach developed by Hammer et al. (2012). This classification approach exploits the abundance of data containing mainly background signals to obtain general wave-field properties. Using these properties, a widespread background model can be learned from the general properties. A new event class is then implemented by using the background model to adjust the event model description by using only one training event. The so obtained classifier therefore consists of the background model and one model for each implemented event class. The classification process itself calculates the likelihood that an unknown data stream has been generated by a specific class for each individual class HMM. More detailed information can be found in Hammer et al. (2012, 2013).

## 4.2 Feature calculation

Although it is possible to use the raw seismic data as input for the hidden Markov model, we used a compressed form of it, so called features. Several features can be calculated representing different aspects of the time series such as spectral, temporal or polarization characteristics. The representation of the seismic signals by features is more adequate to highlight differences





between diverse event types. Since we used single component geophones, we only used spectral and temporal features. Based on preliminary analysis, we used the features listed below.

- – Central frequency

- – Dominant frequency

5 – Instantaneous bandwidth

- – Instantaneous frequency

- – Cepstral coefficients

- – Half-octave bands

A detailed list of the functions used to calculate these features can be found in Hammer et al. (2012).

10 To calculate the features, we used a sliding window with width $w = 1024$ samples. The sliding window is then moved forward with a step of $0.05\,\mathrm{s}$ or 25 samples resulting in an overlap of $97\%$.

For the half-octave bands we used a central frequency of $f_c = 1.3\,\mathrm{Hz}$ for the first band and a total number of 9 bands. Since the geophones have an eigenfrequency of 14 Hz, only signals with a higher frequency are recorded without any loss of information. However, preliminary results showed that half-octave bands with a central frequency higher than $f_{\min} = 5\,\mathrm{Hz}$ are 15 adequate.

### 4.3 Post-processing

The HMM classification resulted in several hundred events in the avalanche class. Many of these events were however of very short duration or only identified at one sensor and did not necessarily coincide with avalanches in the reference catalogue, i.e. these events were likely false alarms. We therefore investigated three post-processing methods to reduce the number of false 20 alarms, namely:

1. Applying a duration threshold for the detected events

2. Analyzing the results of all sensors by introducing a voting based classification

3. Analyzing the coherence between all sensors for each detected event

First, we used an event duration threshold. Based on the analysis of van Herwijnen et al. (2013) and van Herwijnen et al. 25 (2016), we can assume that event duration correlates with avalanche size. The first post-processing method therefore consisted of using a minimal duration $T_{\min}$ for the events. Any automatically detected event $j$ with $T_j \leq T_{\min}$ was thus removed. Similarly, any avalanche $i$ in the reference catalogue with a duration $T_i \leq T_{\min}$ was also removed.

Second, we used a voting based threshold by tallying the classified events of each sensor. Rubin et al. (2012) used a similar approach and found that with increasing votes the false alarm rate decreased. The overall idea is that although an avalanche



event might not be recorded by one sensor, for instance due to poor coupling of the sensor, it is unlikely that an avalanche is missed by all sensors, especially larger avalanches (Faillettaz et al., 2016). Any automatically detected event with $V_j \leq V_{\min}$ with $V$ the number of votes was removed.

Third, we used a threshold based on the cross correlation coefficient between the seven sensors. Wave-fields generated by avalanches should be relatively coherent, while wave fields generated by noise (e.g. wind) are expected to be incoherent. We therefore divided the seismic data in non-overlapping windows of 1024 samples and for each window we defined a mean normalized correlation coefficient

$$R(t_{\mathrm{win}}) = \frac{1}{N_{pairs}} \sum_{k=1}^{N_{pairs}} r_{kl}(t_{\mathrm{win}}) \qquad (2)$$

with $N_{\mathrm{pairs}} = 21$ the number of sensor pairs, $r_{kl}(t_{\mathrm{win}})$ the maximum in the normalized cross correlation between sensor $k$ and $l$ and $t_{win}$ the time of the sliding window. The normalized cross correlation is defined as:

$$\overline{\phi}_{kl}(t) = \frac{\phi_{kl}(t)}{\sqrt{\phi_{kk}(0)\phi_{ll}(0)}} \qquad (3)$$

with $\phi_{kk}(0)$ and $\phi_{ll}(0)$ the zero lag autocorrelation of each sensor which is equal to the energy of each single time window. The maximum of the normalized correlation is picked for a maximum lag of $t_{\max} = 0.05\,\mathrm{s}$, which is the time a sonic wave-field at a speed of $330\,\mathrm{m/s}$ needs to travel the maximum distance between the most distant receiver pair ($\approx 15\,\mathrm{m}$):

$$r_{kl}(t_{\mathrm{win}}) = \max \overline{\phi}_{kl}(t), |t| \leq 0.05\,\mathrm{s} \qquad (4)$$

The normalized cross correlation only yields values between -1 (perfectly anti-correlated) and 1 (perfectly correlated). A value of 0 means that the signals are completely uncorrelated. Finally, the coherence of each automatically detected event was defined as

$$C_j = \max R(t_{\mathrm{win}}), 0 \leq t_{\mathrm{win}} \leq T_j \qquad (5)$$

Any automatically detected event with $C_j \leq C_{\min}$ was removed.

### 4.4 Model performance evaluation

To evaluate the performance of the HMM classification, we compared the automatic picks with the reference data set described in section 3.2. To assign an event classified by the HMM as a positive detection we defined a tolerance interval $d$: the time $t_j^{HMM}$ of an event had to be within the interval $t_i - d \leq t_j^{HMM} \leq t_i + d$, with $t_i$ the release time of the i-th avalanche in the reference data set and $d = 60\,\mathrm{s}$. The tolerance interval $d$ was necessary, since the release times $t_i$ of the avalanches were picked manually and may contain some uncertainties. In addition, the releases times $t_j^{HMM}$ do not necessarily coincide with the reference data since the classifier is not an onset picker.

To describe the performance of the classifier we used three values $N_{\mathrm{hit}}$, $N_{\mathrm{unassigned}}$ and $N_{\mathrm{miss}}$. The first value $N_{\mathrm{hit}}$ describes the total number of avalanches which were correctly detected by the classifier, i.e. events identified by the classifier which

corresponded to avalanches in the reference data set. The second value $N_{\mathrm{unassigned}}$ describes the number of events identified by the classifier which did not correspond to an avalanche in the reference data set. We do not call these events false alarms as during the manual detection some avalanche events might have been missed that are therefore not present in the reference data set. Avalanche events may still be found in the unassigned detections. Finally, the third value $N_{\mathrm{miss}}$ describes the number of avalanches in the reference data that were not identified by the HMM classifier. The three values were used to evaluate the overall model performance in terms of probability of detection (POD) and false alarm ratio (FAR), defined as $POD = \frac{N_{\mathrm{hit}}}{N_{\mathrm{hit}}+N_{\mathrm{miss}}}$ and $FAR = \frac{N_{\mathrm{unassigned}}}{N_{\mathrm{hit}}+N_{\mathrm{unassigned}}}$ (Wilks, 2011).

To determine the best threshold values for the post-processing steps (section 4.3), we plotted POD against FAR values for all probability classes and for different values of $T_{\mathrm{min}}$, $v_{\mathrm{min}}$ and $C_{\mathrm{min}}$. One curve therefore illustrates the POD and FAR with respect to the probability classes for a fixed threshold value. Ideally, when only considering the 100% probability class, the POD value should be 1 while the FAR value should also be relatively high since all detections of the lower classes are counted as false alarms. By taking more probability classes into account, the FAR will decrease while the POD value should stay close to 1. A perfect model should therefore result in constant POD values of 1 whereas the FAR values should decrease to 0. For realistic models, however, POD and FAR values are expected to decrease when accounting for more probability classes. By plotting POD against FAR values for different threshold values, the optimal threshold value can be found by searching for the largest area under the curve (AUC). A perfect model would result in an AUC value of 1, while an AUC value of 0.5 corresponds to a model with a 50-50 chance. Threshold values obtained for models with an AUC value lower than 0.5 are therefore not reliable and any random threshold value can be chosen. In our case, the POD-FAR curves did not span from 0 to 1 on the x-axis. To determine the AUC value we therefore calculated the ratio between the area under the POD-FAR curve and the area under the bisector with the same x-axis limits

$$AUC = \frac{AUC_{\mathrm{hmm}}}{AUC_{\mathrm{bisector}}} \cdot 0.5. \tag{6}$$

## 5 Results

### 5.1 Temporal feature distribution

To investigate changes in feature distribution over time, for instance, due to diurnal changes in environmental noise levels or seasonal changes in snow cover properties, we calculated hourly and daily mean values for all features (see Figure 4 for the central frequency and dominant frequency). Throughout the season there were large variations in the feature distribution at various time scales. First, there were strong diurnal variations (yellow lines in Figure 4). These were observed in all features (not shown). Second, there were also large variations at longer time scales (blue lines in Figure 4). While for some features there were significant seasonal trends, for instance, for the dominant frequency (Pearson $r = -0.56$, $p < 0.001$; Figure 4b), for others there were no clear trends, for instance, for the central frequency (Pearson $r = 0.23$, $p = 0.02$; Figure 4a). Building a single background model to classify the entire season would therefore likely not result in a reliable classification. The background





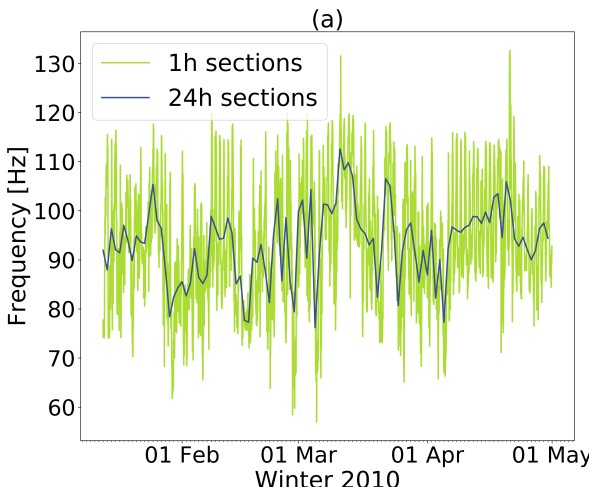
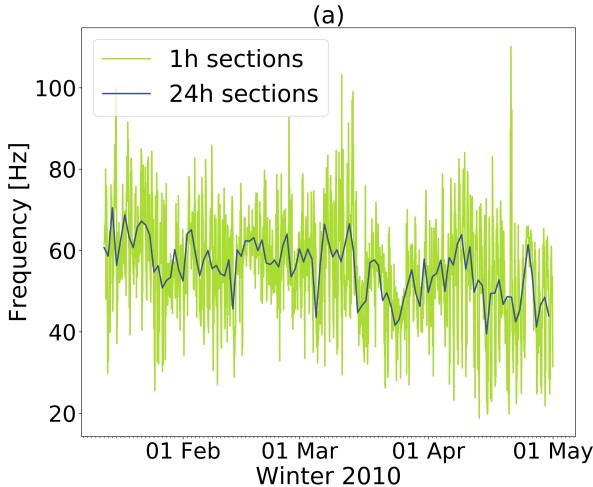

**Figure 4.** (a) Temporal variations in the central frequency hourly average (yellow) and daily average (blue). (b) Temporal variations in the dominant frequency.

model thus has to be regularly recalculated. We therefore decided to recalculate the background model each day to classify the events within the same day.

## 5.2 Training event

While building a representative background model is important, choosing an appropriate training event is of utmost importance
for the classifier. As outlined in van Herwijnen et al. (2016), the avalanche catalogue consists of various avalanches of different
size and type. At the beginning of the season the avalanches are most likely dry-snow avalanches, while at the end of the
season there a mostly wet-snow avalanches. Our avalanche catalogue only consists of the release time and little information is
available on the type of the avalanches. We therefore compared the feature distribution of four different avalanche events, of
21 January, 27 February, 22 March and 24 April 2010 to investigate whether substantial differences related to avalanche type
existed (Figure 5).

While there were some subtle differences in the feature distribution for the four avalanches (e.g. between the avalanches
on 21 January 2010 and 22 March 2010 in Figure 5 (a)), overall the four avalanches exhibited very similar behaviour. Thus,
when viewing avalanches in the feature space, wet- and dry-snow avalanches appear to be very similar. We therefore used
one single avalanche class for the HMM classifier and used one training event to learn the model. Specifically, we used an
avalanche with $P_{\mathrm{ava}} = 100\%$ recorded on 22 March 2010 (Figure 6) with a duration of $30\,\mathrm{s}$. However, as seen in Figure 5, the
most rapid changes in feature values occurred at the beginning of the events. In the coda changes in feature values were rather
slow, providing limited relevant information for the classifier. We therefore only used the first 8 s of the training event (marked
by the red rectangle in Figure 6).





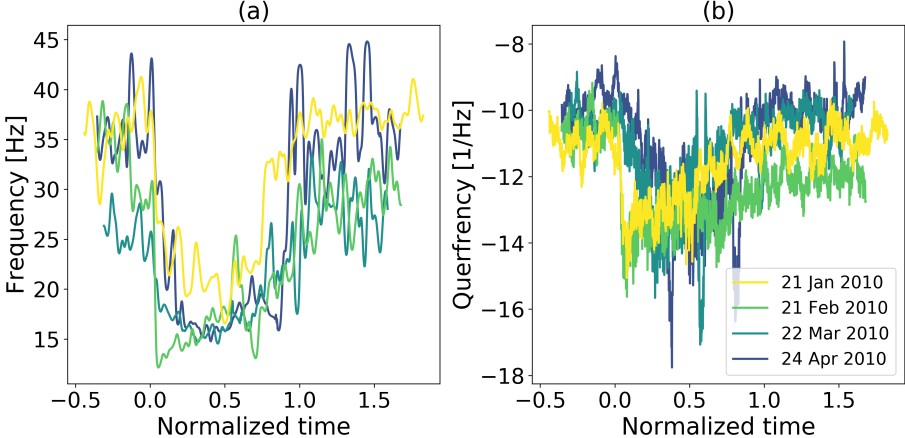

**Figure 5.** (a) Central frequency with normalized time for 4 different avalanche events (colours) from 4 different months. For comparison, the time was normalized by the event duration with 0 indicating the start of the avalanche and 1 the end of the event. (b) $2^{nd}$ cepstral coefficient.

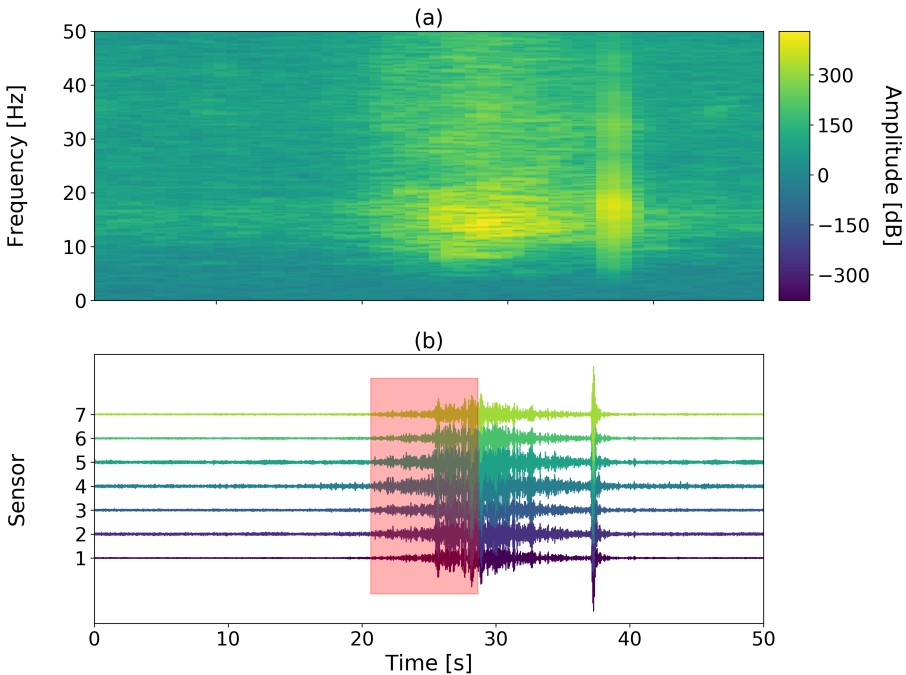

**Figure 6.** The event used for training the HMM. (a) Stacked spectrogram of all 7 sensors. (b) Seismic waveform for each individual sensor (colours). The red area highlights the part of the signal used as the training event for the HMM.





**Table 2.** Number of detections ($N_{\mathrm{hit}}$) and probability of detection (POD) for each sensor as well as the number of events that were not in the reference data set ($N_{\mathrm{unassigned}}$).

| Probability Class | | 0% | 16.6% | 33.3% | 50% | 66.6% | 83.3% | 100% | $N_{\mathrm{unassigned}}$ |
|---|---|---|---|---|---|---|---|---|---|
| Sensor 1 | POD | $\frac{12}{58}$ | $\frac{17}{66}$ | $\frac{21}{48}$ | $\frac{17}{34}$ | $\frac{14}{30}$ | $\frac{21}{27}$ | $\frac{19}{20}$ | 149 |
| | | 21% | 26% | 44% | 50% | 47% | 78% | 95% | |
| Sensor 2 | POD | $\frac{28}{58}$ | $\frac{24}{66}$ | $\frac{19}{48}$ | $\frac{11}{34}$ | $\frac{17}{30}$ | $\frac{17}{27}$ | $\frac{14}{20}$ | 1432 |
| | | 48% | 36% | 40% | 32% | 57% | 63% | 70% | |
| Sensor 3 | POD | $\frac{21}{58}$ | $\frac{25}{66}$ | $\frac{21}{48}$ | $\frac{12}{34}$ | $\frac{16}{30}$ | $\frac{17}{27}$ | $\frac{15}{20}$ | 1347 |
| | | 36% | 38% | 44% | 35% | 53% | 63% | 75% | |
| Sensor 4 | POD | $\frac{20}{58}$ | $\frac{19}{66}$ | $\frac{18}{48}$ | $\frac{9}{34}$ | $\frac{15}{30}$ | $\frac{15}{27}$ | $\frac{14}{20}$ | 2091 |
| | | 35% | 29% | 38% | 27% | 50% | 56% | 70% | |
| Sensor 5 | POD | $\frac{22}{58}$ | $\frac{27}{66}$ | $\frac{17}{48}$ | $\frac{13}{34}$ | $\frac{19}{30}$ | $\frac{17}{27}$ | $\frac{15}{20}$ | 786 |
| | | 38% | 41% | 35% | 38% | 63% | 63% | 75% | |
| Sensor 6 | POD | $\frac{11}{58}$ | $\frac{11}{66}$ | $\frac{16}{48}$ | $\frac{14}{34}$ | $\frac{15}{30}$ | $\frac{20}{27}$ | $\frac{18}{20}$ | 124 |
| | | 19% | 17% | 33% | 41% | 50% | 74% | 90% | |
| Sensor 7 | POD | $\frac{29}{58}$ | $\frac{26}{66}$ | $\frac{23}{48}$ | $\frac{16}{34}$ | $\frac{19}{30}$ | $\frac{17}{27}$ | $\frac{14}{20}$ | 2094 |
| | | 50% | 39% | 48% | 47% | 63% | 63% | 70% | |

## 5.3 Automatic avalanche classification

### 5.3.1 Single sensor classification

We used the entire reference data set containing 283 avalanche events to evaluate model performance as function of probability class.

The first model was built for each individual sensor and without any post-processing. For each sensor a separate model was built containing only the data of the specific sensor. In Table 2 the number of detections for each probability class is listed. For the highest probability class the POD values were relatively high, ranging from 70% to 95% and the values generally decreased with decreasing probability class. Nevertheless, even for the lowest probability class events were still detected. Furthermore, numerous events, between 124 and 2091 events, were detected for each sensor which were not listed in the reference data set.

Clearly without any post-processing the number of unassigned events is high.

To reduce the number of unassigned events we applied a minimum duration $T_{\mathrm{min}}$ to remove events. To obtain a reasonable threshold value, we determined the area under the POD-FAR curve for different $T_{\mathrm{min}}$ values (Figure 7).

Due to the large number of unassigned events (see Table 2) AUC values were generally below 0.5 and using a minimum duration threshold did not result in much improvement. However, for sensors 1 and 6 the number of unassigned events was

much lower and there was an optimum $T_{\mathrm{min}}$ threshold value around 12 s (Figure 7 (b)). In the following, we therefore used the same minimal duration $T_{\mathrm{min}} = 12\,\mathrm{s}$ for all sensors. While with this $T_{\mathrm{min}}$ threshold the POD values for the highest probability





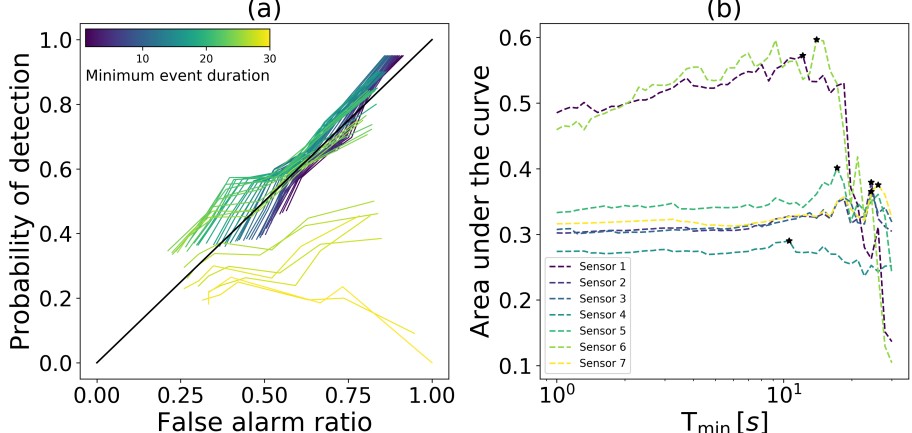

**Figure 7.** (a) POD-FAR curves for sensor 1 for different minimum event durations $T_{\mathrm{min}}$ (colours). (b) Area under the POD-FAR curve with $T_{\mathrm{min}}$ for all sensors (colours). The stars show the $T_{\mathrm{min}}$ value with the largest area under the curve for each sensor.

**Table 3.** Number of detections ($N_{\mathrm{hit}}$) and probability of detection (POD) for each sensor as well as the number of events that were not in the reference data set ($N_{\mathrm{unassigned}}$) using a minimal event duration of $T_{\mathrm{min}} = 12\,\mathrm{s}$.

| Probability classes | | 0% | 16.6% | 33.3% | 50% | 66.6% | 83.3% | 100% | $N_{\mathrm{unassigned}}$ |
|---|---|---|---|---|---|---|---|---|---|
| Sensor 1 | POD | $\frac{0}{0}$ | $\frac{4}{39}$ | $\frac{6}{36}$ | $\frac{11}{23}$ | $\frac{10}{26}$ | $\frac{15}{27}$ | $\frac{16}{19}$ | 22 |
| | | 0% | 10% | 17% | 48% | 39% | 56% | 84% | |
| Sensor 2 | POD | $\frac{0}{0}$ | $\frac{17}{39}$ | $\frac{12}{36}$ | $\frac{7}{23}$ | $\frac{14}{26}$ | $\frac{17}{27}$ | $\frac{14}{19}$ | 408 |
| | | 0% | 44% | 33% | 30% | 54% | 63% | 74% | |
| Sensor 3 | POD | $\frac{0}{0}$ | $\frac{14}{39}$ | $\frac{13}{36}$ | $\frac{9}{23}$ | $\frac{13}{26}$ | $\frac{16}{27}$ | $\frac{14}{19}$ | 340 |
| | | 0% | 36% | 36% | 39% | 50% | 59% | 74% | |
| Sensor 4 | POD | $\frac{0}{0}$ | $\frac{7}{39}$ | $\frac{13}{36}$ | $\frac{5}{23}$ | $\frac{10}{26}$ | $\frac{14}{27}$ | $\frac{13}{19}$ | 597 |
| | | 0% | 18% | 36% | 22% | 39% | 52% | 69% | |
| Sensor 5 | POD | $\frac{0}{0}$ | $\frac{16}{39}$ | $\frac{11}{36}$ | $\frac{9}{23}$ | $\frac{15}{26}$ | $\frac{14}{27}$ | $\frac{13}{19}$ | 157 |
| | | 0% | 41% | 31% | 39% | 58% | 52% | 68% | |
| Sensor 6 | POD | $\frac{0}{0}$ | $\frac{3}{38}$ | $\frac{5}{35}$ | $\frac{6}{22}$ | $\frac{8}{26}$ | $\frac{12}{26}$ | $\frac{15}{19}$ | 15 |
| | | 0% | 8% | 14% | 27% | 31% | 46% | 79% | |
| Sensor 7 | POD | $\frac{0}{0}$ | $\frac{17}{39}$ | $\frac{14}{36}$ | $\frac{10}{23}$ | $\frac{15}{26}$ | $\frac{17}{27}$ | $\frac{14}{19}$ | 653 |
| | | 0% | 43% | 39% | 44% | 58% | 63% | 74% | |

class somewhat decreased, they still remained high (between 68% and 84%; Table 3). Note that the duration threshold was also applied to remove events from the reference data set. For a threshold value of $T_{\mathrm{min}} = 12\,\mathrm{s}$, the number of events in the reference data set reduced from 283 to 170.





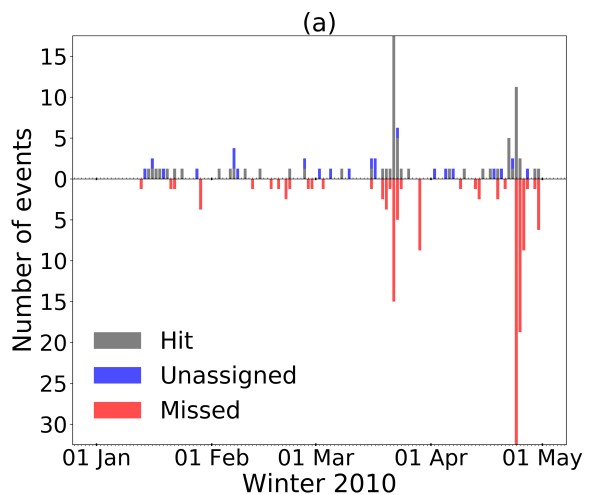
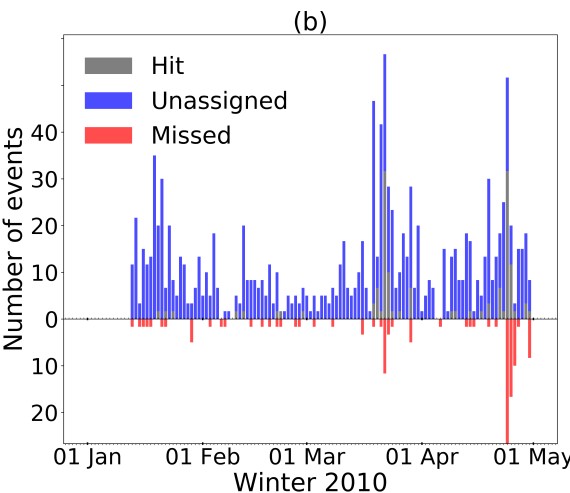

**Figure 8.** Classification results using a minimal event duration length of $12\,\mathrm{s}$ for (a) sensor 1 and (b) sensor 7. Gray bars show the number of matching detections (hit), blue bars show the unassigned events and the red bars show the events that were not detected as an avalanche (missed).

Overall the number of unassigned events substantially decreased (compare Tables 2 and 3), especially for sensors 1 and 6. For these two sensors the POD values also substantially decreased for the lower probability classes but the two main avalanche periods in March and April are clearly visible in the detections (see Figure 8a for sensor 1). However, for the other sensors, the POD values remained relatively high for the lower probability classes and there were still many unassigned events and the two main avalanche periods were less evident (see Figure 8b for sensor 7). Clearly, for some of the sensors the number of unassigned events remained very high.

### 5.3.2 Array based classification

To further reduce the number of unassigned events (potential false alarms), we applied two array based post-processing methods to eliminate events, namely a minimum number of votes and coherence threshold (see Section 4.3). To define an optimal number of votes or coherence threshold we used the same procedure as for the determination of a minimal duration threshold (Figure 7). Since for these array based methods the detections from all sensors were pooled, the number of unassigned events was very high. This resulted in much higher FAR values and thus poor model performance with low AUC values, all below 0.5. Arbitrary values for $v_{\min}$ or $C_{\min}$ could therefore be chosen. However, the overall goal of these post-processing steps was to reduce the number of unassigned events, while still retaining a reasonable probability of detection. We therefore analysed the effect of different threshold values on FAR values as well as on POD values for the probability classes.

Overall, for both processing steps the number of unassigned events decreased with increasing threshold values (Figures 9 and 10). By using a minimal number of 5 votes the POD stayed relatively high with a low number of unassigned events. Following




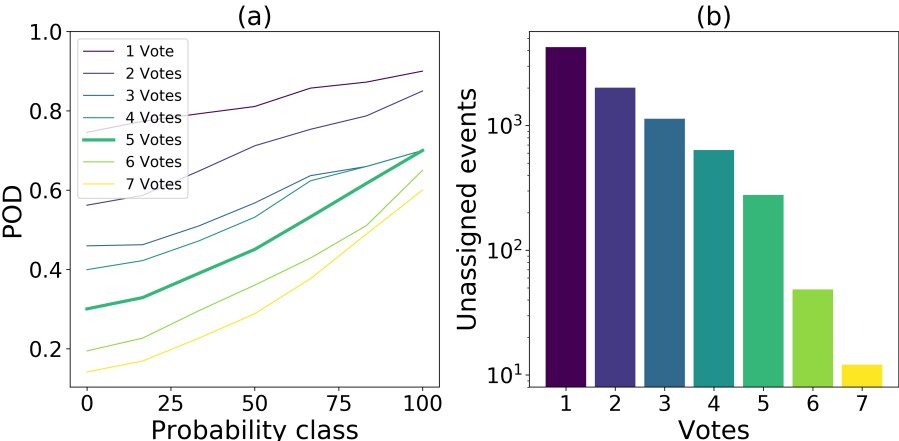

**Figure 9.** (a) POD for each probability class depending on the minimum number of votes (colours). (b) Number of unassigned events for different number of votes (colours).

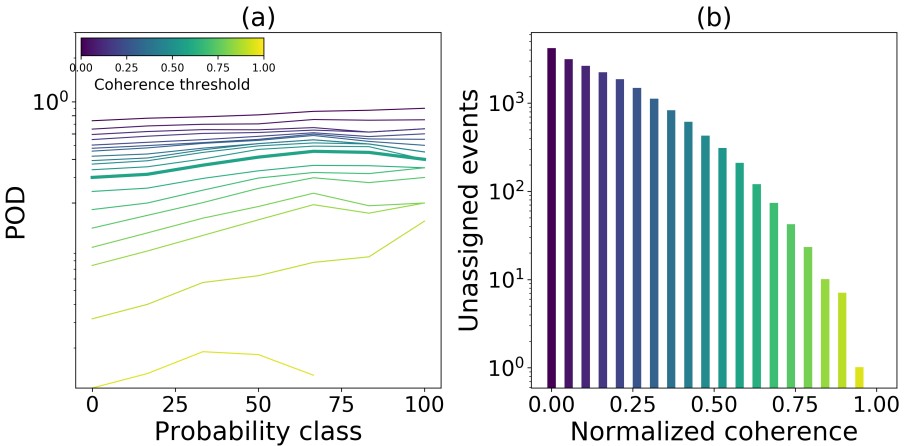

**Figure 10.** (a) POD for each probability class depending on the minimal coherence (colours). (b) Number of unassigned events for different coherence threshold values (colours).

the same procedure we selected a coherence value of 0.6. Using this value, the POD was relatively high and the number of unassigned events could be reduced to less than 100 events.

In total, we thus have three different post-processing steps which can be applied to the data: a minimal event duration $T_{\min} = 12\,\text{s}$, a minimum number of votes $v_{\min} = 5$ and a coherence threshold $C_{\min} = 0.6$. By combining these three steps, six

5  array based post-processing work flows were implemented:

   − $v : v_j \geq v_{\min}$





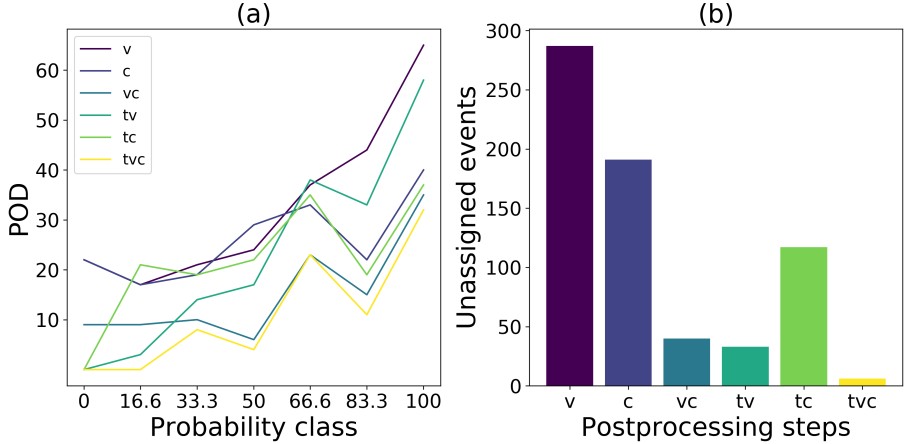

**Figure 11.** (a) POD for each probability class for different post-processing work flows (colours). v = minimal number of votes used, t = minimal event duration and c = minimal coherence. (b) Number of unassigned events remaining after the post processing for each work flow.

- $c : C_j \geq C_\mathrm{min}$

- $vc : v_j \geq v_\mathrm{min}$ and $C_j \geq C_\mathrm{min}$

- $tv : T_j \geq T_\mathrm{min}$ and $v_j \geq v_\mathrm{min}$

- $tc : T_j \geq T_\mathrm{min}$ and $C_j \geq C_\mathrm{min}$

- $tvc : T_j \geq T_\mathrm{min}, v_j \geq v_\mathrm{min}$ and $C_j \geq C_\mathrm{min}$

The number of unassigned events decreased most with a combined approach always including the number of votes (vc, tv, or tvc in Figure 11). When using only one array based post-processing step, the number of unassigned events remained high (v and c in Figure 11b). While the lowest number of unassigned events was achieved when combining all three post-processing steps, this model also resulted in low POD values for all probability classes (tvc Figure 11a). Overall, the highest POD values
and the steepest decrease for the lowest probability classes were obtained for the voting based processing with and without a minimal duration of the events ($v$ and $tv$ in Figure 11a). For both these post-processing work flows the two periods of high avalanche activity are visible (Figure 12). However, by also applying the duration threshold, the total number of detections decreased and the two periods in March and April became more clearly visible (Figure 12b).

### 5.3.3   Unassigned events

We compared the automatic detections with the reference avalanche catalogue and obtained a large number of unassigned events. It remains unclear whether the unassigned events are all false alarms or partly correspond to avalanches that were not identified in the reference data. We therefore visually inspected the unassigned events. In order to keep this reanalysis



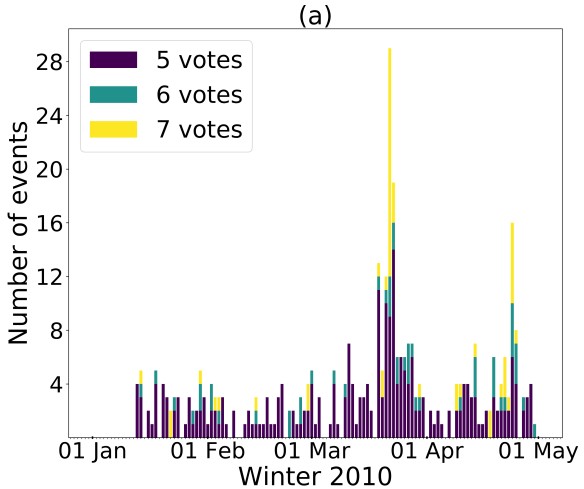 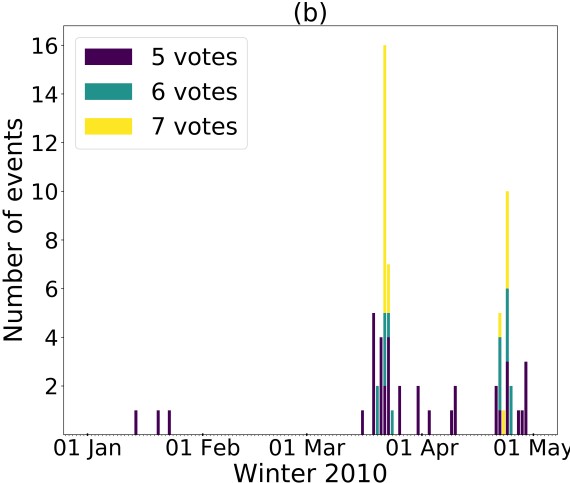

**Figure 12.** All events detected by at least 5 sensors. The different colours indicate the number of votes. (a) the results without any limitation of the duration of the events are plotted, (b) only events with a minimum duration as mentioned before are taken into account. On the right the two main avalanche periods are more clearly visible.

**Table 4.** Results of the reanalysis of the detections not covered by the test data set. Left side of the table shows the results of the reanalysis of two single sensors, the right side shows the results of the array based classification. For the single sensors a minimum duration for the events of $t_{min} = 12s$ was taken into account. The voting based processing steps analysed are minimum number of votes (v), minimum duration (t) and coherence (c).

|  | Single sensors | | Array based | | |
|---|---|---|---|---|---|
|  | Sensor 1 | Sensor 6 | tv | vc | tvc |
| Unassigned | 22 | 15 | 33 | 40 | 6 |
| Confirmed as avalanche | 36% (8) | 47% (7) | 12% (4) | 28% (11) | 33% (2) |
| Confirmed as false alarm | 27% (6) | 27% (4) | 79% (26) | 45% (18) | 50% (3) |
| Uncertain | 36% (8) | 27% (4) | 9% (3) | 28% (11) | 17% (1) |

manageable, we only focused on the post-processing steps which resulted in less than 50 unassigned events. Thus, we only investigated single sensor results from sensor 1 and 6 with $T_{min} = 12\,\mathrm{s}$ and array based results for tv, vc and tvc. The visual inspection showed that for the single sensor results between 36% and 47% of the unassigned events were likely unidentified avalanches and less than one third were false alarms (Table 4). For the array based results, however, most of the unassigned events (between 45% and 79%) were false alarms while fewer events were likely associated with avalanches that were missed by van Herwijnen and Schweizer (2011a).



## 6   Discussion and Summary

We trained hidden Markov models (HMMs), a machine learning algorithm, to automatically detect avalanches in continuous seismic data recorded near an avalanche starting zone above Davos, Switzerland for the winter season of 2010. To reduce the amount of data to process, we pre-processed the continuous data using an amplitude threshold (Figure 2). We then implemented

single sensor and array based post-processing steps and the performance of the models was evaluated using a previously published reference avalanche catalogue obtained from the same seismic data (van Herwijnen and Schweizer, 2011a; van Herwijnen et al., 2016).

After pre-processing the data, the reference avalanche catalogue contained 283 avalanches between 12 January and 30 April 2010, events that were identified by visual inspection of the waveform and spectrogram of a single sensor (van Herwijnen and

Schweizer, 2011a). Since only 33 of these events were independently confirmed avalanches, considerable uncertainty remained about the identified events. To reduce the uncertainty in the reference catalogue, three of the authors therefore re-evaluated the data. This allowed us to assign 7 subjective probability classes between 0% and 100% to each event. Overall, only 20 events were marked as certain avalanche (Table 1) and hence the performance, of the classifiers can only be evaluated for these particular events. For the remaining events, there are still uncertainties and hence the performance of the classifier can only

be estimated. Furthermore, this reanalysis highlighted the difficulty in obtaining an objective and reliable reference avalanche catalogue. It also showed that expert decisions are biased and there is a need for a reliable automatic classifier to identify avalanches in continuous seismic data.

Recent work by Hammer et al. (2017) showed very promising results for applying a HMM to automatically detect avalanches in continuous seismic data. While they only focussed on a five day period during an exceptional avalanche cycle in 1999, our

goal was to classify continuous seismic data spanning more than 100 days. This prevented us from building a single background model to classify the entire season since temporal variations in feature distributions at various time scales were present (Figure 4). Indeed, when using a single background model to classify the entire season for sensor 1 only two thirds of the events were detected by having almost 6 times the number of unassigned events. One possible reason for these variations in feature distribution was likely the setup of the sensor array. The geophones were packed in a Styrofoam housing and inserted within the

snowpack. As such, less snow covered the sensors than if they has been inserted in the ground, making them more susceptible to environmental noise. Furthermore, it is also likely that the snow cover introduced additional noise in spring due to the rapid settlement and water infiltration. We therefore recalculated the background model for each day and for each sensor to classify the data from the same day. However, for the operational implementation this would be impractical, since there would always be a 24 hour delay in the detections. Other strategies for regularly updating the background model should therefore be

investigated (e.g. Riggelsen and Ohrnberger, 2014).

We performed the automatic classification over the entire season by recalculating the classifier for each day and for each sensor. Overall, probability of detection (POD) values decreased with decreasing probability class and the highest POD values were associated with the highest probability class for all sensors (Table 2). Indeed, between 70% and 95% of all avalanches in the highest probability class were detected, which is comparable to the results presented by Bessason et al. (2007) and





Leprettre et al. (1996), who reported POD values of approximately 65% and 90%. Nevertheless, without any post-processing, the number of unassigned events was high, questioning the reliability of the models as many of these events were likely false alarms. Post-processing of the results was therefore required. Applying a minimal signal duration drastically reduced the number of unassigned events while still retaining reasonable POD values, in particular for sensor 1 and sensor 6 (Table 3). However, there were large differences in model performance between the sensors (Figures 7 and 8). The reason for these performance differences is very likely the deployment of the sensors. Indeed, sensor 1 and 6 were deployed at the top of the slope closest to a cornice where the snow was the deepest (van Herwijnen and Schweizer, 2011a). The other 5 sensors were covered by less snow due to local inhomogeneities, leaving these sensors more sensitive to environmental noise. For future deployments it will thus be important to deploy the sensors below a homogeneous snow cover and not within the snow cover. This should reduce the amount of environmental noise and consequently the number of false alarms.

To further reduce the number of false alarms, we implemented two array based post-processing steps, namely a voting based approach and a signal coherence threshold. In combination with the minimal event duration, we thus investigated 6 array based post-processing work flows. Results showed that these array based methods were effective in reducing the number of unassigned events (Figure 11). However, the POD values generally also decreased, resulting in overall fewer detections. Combined post-processing methods which included the voting based approach resulted in better model performance, in line with results presented by Rubin et al. (2012). The best model performance was obtained by combining the event duration threshold for events with at least 5 votes. The number of unassigned events reduced to about 30 and POD values were highest ($\sim$ 55%) for the highest probability class and decreased for the lower classes. Despite the large differences in model performance for the individual sensors, the model still performed marginally better when pooling the data from the entire array. These results are promising as with an improved sensor deployment strategy array based post-processing is likely to further improve.

Comparing our model performance to previously published studies is not straightforward. We assigned subjective probability classes to our reference avalanche catalogue rather than using a yes or no approach. Furthermore, we used geophones deployed in an avalanche starting zone, while Bessason et al. (2007), Leprettre et al. (1996) and Hammer et al. (2017) used sensitive broadband seismometers deployed at valley bottom. Therefore, it is very likely that there was more environmental noise in our data and many of the detected avalanches in our reference data set were rather small (van Herwijnen et al., 2016). Given these differences in instrumentation and deployment, our detection results are encouraging and highlight the advantage of using HMMs for the automatic identification of avalanches in continuous seismic data. Indeed, our model only required one training event (Figure 6) to classify the entire season. As shown by Hammer et al. (2017), for large avalanches it is possible to build a HMM with a high POD and very low FAR with one training event. As such, HMMs are well suited to detect avalanches as they can easily be implemented at new sites. In contrast, the model used by Bessason et al. (2007) relied on a 10-year data base, and Leprettre et al. (1996) used a set of fuzzy logic rules derived by the experts. Note that the post-processing steps we investigated are likely site-dependent, in particular the event duration threshold. However, such a threshold value is intuitive, has a linear influence on model outcome and is thus easily tunable.

Overall, our results suggest that HMMs may be well suited for the automatic detection of avalanches in a continuous seismic data for operational avalanche forecasting. The variable model performance between the different sensors highlighted some





problems which can likely be overcome by improving the sensor deployment strategy. Specifically, we suggest that the sensors should be deployed 30 to 50 cm underground at a site with a homogeneous and preferably thick snow cover. Furthermore, the distance between the sensors should be increased to apply array processing techniques for source localization (Lacroix et al., 2012). Finally, incorporating localization parameters as new features in the HMM could open the door for further model

5  improvement, as is done for the automatic detection of avalanches in continuous infrasound data (Marchetti et al., 2015; Thüring et al., 2015).

*Acknowledgements.* M.H. was supported by a grant of the Swiss National Science Foundation (200021_149329). We thank numerous colleagues from SLF for help with field work and maintaining the instrumentation.

*Competing interests.* The authors declare that they have no conflict of interest.





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
