# Peer review of "Automatic detection of snow avalanches in continuous seismic data using hidden Markov models"

_Natural Hazards and Earth System Sciences, 2017_

## Referee Comment (RC1) · Anonymous Referee #1 · 20 Jul 2017

This study focuses on the seismic detection and identification of the signals generated by snow avalanches in the Davos area in the Swiss Alps during the winter 2010. The authors tested the capability of a machine learning algorithm (hidden Markov models - HMM) to perform this detection and identification from continuous seismic data. They used a reference catalog to evaluate the performance of the algorithm. The first results showed that the algorithm is capable to achieve relatively high positive identification rates of the avalanches in the catalog (70-95% depending on the station that recorded the signals), but also with a high rate of supposedly false detections. This led the authors to propose a post-processing strategy. Three post-processing steps were investigated: (i) analysis of the duration of the signals; (ii) computation of a correlation

factor to evaluate the coherence of the signal between each sensor; and (iii) a voting system based on the classification returned by each station for a given event. Using one, or a combination of those proposed post-processing steps, led to a decrease of false alarm rates, but also in most cases to a decrease of the rate of good identification.

The use of seismology to study environmental processes is of growing interest as it allows producing observations with a unique spatio-temporal resolution. This new approach can help to better understand the triggering factors of natural hazards and to mitigate their consequences on our societies. In this context, this study contributes to the continuing effort to develop robust and versatile methods to explore years of continuous data and for the implementation of real-time seismology-based warning systems. Overall, I think the paper is clearly written, and that the Authors have made a good effort to carefully explore the data, explain their approach and discuss their results. Nevertheless I listed below several comments and suggestions that might help to improve this paper.

General comment:

The only major concern I have regarding this work is that the downsides of using the HMM algorithm are not discussed while most of the results presented in this paper suggests that HMM alone, without pre- and post-processing, cannot perform identification of seismic sources with a high success rate. The strengths of the HMM are usually stated to be: i) it does not need any pre-detection or picking (STA/LTA, etc.), which should ensure that no event is missed; ii) it does not require any inputs from experts. Yet this paper demonstrates that i) a pre-detection can be suitable to remove low-amplitude/noise signals (figure 2); ii) post-processing steps with thresholds set by experts (duration, etc.) is needed to achieve a high accuracy.

Moreover, the Authors are building their post-processing strategy based on features that can be incorporated in the identification models constructed with other machine learning algorithms. The post-processing steps the Authors propose seem necessary
because the HMM cannot include these features (durations of the signals, coherence between signals recorded at different stations, vote among stations) in the model due to its core design, which is to consider chunk of continuous data and not the entire signal generated by the event. This forces the Authors to manually set thresholds on those features, while with some other algorithms those thresholds are determined through a statistical analysis of the reference data.

I think the authors must include a more thorough and objective discussion on the pros (which definitely exist) and the cons of HMM compared to other algorithms/studies in the light of the results of this work.

Specific comments:

P3 L6-8: Are those false alarm rates related to the choice of the algorithms or to the choice of the features used to parametrized the signals? The latter might be more important and should be mentioned.

Figure 1: I think a colorscale with more colors would allow to better observe the features of the signals generated by the different sources, especially at frequencies below 50 Hz. This is important as the readers might want to understand what guided your choice of features. Also this figure can be larger.

P9 L25-27: How do you compute the duration?

P9 L1-3: How does the voting step in the post-processing would impact the detection of "small" events (especially with a threshold set at 5 stations for a network with 7 sensors)? Are "small" events detected by the whole network? A figure showing the locations of the avalanche corridors and the seismic network would be interesting.

P11 L4-5: Indeed. How would it have impacted your results if you had chosen another master event? Is this has been investigated in Hammer et al. (2017)? If yes it should be mentioned and referenced.

P13 L11-12: So the selection of the threshold on the duration is not done by consid-
ering the physics of the sources or the distribution of the durations in the reference catalog, but to optimize the POD-FAR ratio? By applying this threshold to the reference data set you lose 40% of the events (P14, last line). How is this threshold choice impacting the detection of "small" avalanches? Can you show a histogram of the duration of the events in the reference catalog? You state on P3 L24 that "For avalanche forecasting information on smaller avalanches is also required". Hence is the approach you propose suitable to detect the smaller events?

P13 L14-16, P20 L6-7, Table 3: Again, it would be great to have a map of the seismic network to discuss the discrepancies observed at the different sensors. Distance to the sources, travelled paths, attenuation, dispersion, etc., can also be factors impacting the amplitude, the duration and more generally the features of the signals that might in return change the POD at different stations. This could be discussed.

P20 L17-18: So in the best case what is your overall accuracy? Considering which range of avalanche sizes? I think it is this information that the readers will seek.

P20 28-30: Are the results presented in this study supporting this statement or is it based solely on the study by Hammer et al. (2017)?

P21 L4-5: How can you incorporate localization parameters in the HMM? Is this done directly in the model or during the post-processing?

---

## Referee Comment (RC2) · N. Vouillamoz (Referee) · 17 Aug 2017

General comments:

The paper discusses the performance of an automatic seismic detection system of snow avalanches using hidden Markov models (HMMs). The study is based on a 107-day continuous dataset acquired during the winter 2010 by a small seismic array consisting of 7 vertical geophones deployed above Davos, Switzerland, and surrounded by several avalanches starting zones. The HMMs system is tested against a reference avalanche catalogue that is based on the work by van Herwijnen and Schweizer (2011a). The reference avalanche catalogue gathers 283 events distributed into 7 probability classes following an independent re-evaluation of the avalanche signals of van Herwijnen and Schweizer (2011a) by three of the authors. Among the 283 events, only 20 (7%) are finally classified by the aforementioned three authors as being with 100% probability a snow avalanche. For the HMMs detector, the authors apply the approach developed by Hammer et al. (2012) that requires a background noise model and only one training event to learn the system. Because important diurnal and seasonal variations are observed in the seismic feature of the background noise over the season, a background noise model is recalculated for each day. Since the authors observe that the change from dry to wet avalanches through the winter does not have a strong impact on the selected seismic features used for the HMMs, only one avalanche signal is selected as a training event for a single avalanche class HMMs detector. The first HMMs avalanche detector results in high probability of detection (POD 70-95%) in the 100% probability class of the reference avalanche catalogue for all 7 stations. However, a large number (up to 2091) of additional events (unassigned detections) are detected by the HMMs for each sensors. In order to reduce the number of unassigned detections, the authors decide to introduce post-processing steps to the HMMs: (i) a duration based classification and (ii) an array based (multiple-station detection/coherence of signal among station pairs) classification. The post-processing steps reduce the number of unassigned detections by the HMMs; however, the POD in the highest probability class of the reference avalanche catalogue is also reduced. A manual re-evaluation of the unassigned detections after post-processing shows that part of these detections correspond to avalanche missed during the manual detection procedure.

The paper is well structured and clearly written. Despite the noisy character of the data, the authors show HMMs as developed by Hammer et al. (2012) to be a promising approach for operational avalanche forecasting; however, with fine tuning. Strategies will be needed for a regular update of the background noise model and post-processing must be implemented. Much attention should be paid to the stations deployment. The author recommend to install the sensors 30-50 cm in the ground below a homogenous snow cover in order to reduce environmental noise and increase coherence of the data

among the network and thus, stabilize the array based post-processing. In addition, a larger sensor inter-spacing would allow the system to apply array-processing techniques and enable to incorporate localization parameters of avalanche source area in the automatic system. In my opinion, the implementation of the post-processing step is of great interest, as it helps to deal with the noise contamination that is found at any surface site data. The main concern I have is that the authors didn't make use of the 33 confirmed avalanches of van Herwijnen and Schweizer (2011a) and used instead a re-processed probabilistic reference catalogue where only 20 avalanches remain considered as certain. In the following, a number of specific comments are related to this concern. I leave it to the authors to decide to which point they want to integrate these suggestions. It would be interesting to know how the 33 confirmed events match the post-processing steps: do they have durations above 12 seconds, are they detected by 5 sensors or more, what are the inter-station correlation indices for these events. In addition, if the location of the 33 confirmed events is known, how important are the variations in the spectral feature of the signals as a function of station-source distance? Finally, what is the performance of the HMMs system including the three post-processing steps against this original 385 events/33 confirmed events reference catalogue?

Specific comments:

P1 L9-12. Please rephrase and clarify this part of the abstract (according to remarks about P4-7 Section 3).

P4 L5-7. Although the station inter-spacing is specified in van Herwijnen and Schweizer (2011b), you should mention it here, so the reader has an idea of the size of the network. Please specify also the station numbers, in particular the geophone installed into the ground.

P4-7. I think the section 3 needs some clarification so the reader can get a better idea of the reference data:

СЗ

1) What are the exact data that were analyzed visually by van Herwijnen and Schweizer (2011a) (385 detection, 33 confirmed events): (1) 107-day of continuous data without pre-processing of section 3.1 at sensor number X or (2) XX% of pre-processed data (section 3.1) at sensor number X? In case (1), I would place P4 L28-30 before section 3.1 for clarity.

2) Section 3.1: The pre-processing of the data is obscure to me. Was it apply only to reduce the amount of data to (re-)process manually? Did van Herwijnen and Schweizer (2011a)/Heck et al. AND the HMMs perform only on the 20% of high amplitude data (which would impact on the paper title as it would be no more continuous data)? If a local network is intentionally deployed to detect small avalanches, why place an energy threshold in the investigated data series? See for example avalanche number 5 (Av 5) in Figure 7 of van Herwijnen and Schweizer (2011b) which has poor SNR. Furthermore, HMMs are not dependent on energy thresholds, please clarify.

3) Section 3.2: 20 merged avalanches represent 7% of the 283 reference events. In these data, where are the 33 confirmed avalanches of P4 L30? You could have used 33 confirmed events (9%) against 352 uncertain events of the original 385 events of van Herwijnen and Schweizer?

4) P6 L11-15: I think it would be interesting to discuss here in a few words the 283 events reference avalanche catalogue in more details. For example, do you find trends in the probability classes, are the 20 certain events confirmed detections, long duration signals, high energy events detected by all stations? Are the lower confidence event simply lower quality events (low SNR, only recorded at a few stations and not consistent among the stations)? These low probability events represent in my opinion the real challenge of environmental seismology case studies. And I expect the subjectivity of the analyst to have more influence on low quality data (how would/did you rate avalanche Av 5 in Figure 7 of van Herwijnen and Schweizer (2011b)) than on high SNR data? This is where automatic, quantitative systems should help :)

5) P7 Figure 3. How does the original catalogue with 385 events, 33 confirmed avalanches plot? Are the two periods of higher activity well represented? I personally would insert a plot of the 33 confirmed events at the bottom of Figure 3.

P10 L8-20. Does this very sophisticated approach to find a threshold value bring more than an approach using common sense (for which arguments are well described in P8 L24-29 – P9 L1-20)? The minimum duration can be evaluated (or decided) as a function of the expected distance/size of the target events; the number of stations that must detect the signal can be selected as a reasonable value knowing the network/stations specific performances; the inter-station correlation threshold can be evaluated by investigating the 33 confirmed events. Please comment.

P11 L11-13. I agree that in Figure 5a the feature distribution is very similar for the 4 selected events. However, in Figure 5b, I find events of 21 Jan/21Feb dissimilar to events of 22Mar/24Apr, especially at the events onset where one group goes up while the other one goes down. Please comment on that and see remark on P15 Figure 8.

P12 Figure 5. In (a) and (b) vertical dotted lines at normalized time 0 and 1 would help to visualize start and end time of event.

P15 Figure 8. For both sensor 1 and 7, the HMMs missed 4 events at the end of January:

1) I don't find this 4-event spike in Figure 3?

2) Are any of the 33 confirmed avalanches found at this date? I see there is one 100% probability class event around this date in Figure 3.

3) Could this speak for an influence of the HMMs system by the avalanche type (drywet) selected to train the system?

P18 L1-6. Clarification for the single sensor results:

1) Did you investigate the signals of the unassigned detections individually at station

1 and 6? Or, 2) Did you look at the signals recorded by all stations at the time of the unassigned detections of sensor 1 and 6?

I think these results should be commented. What could explain the higher false alarm rate of the array based approach against the single sensor approach?

P19 L8-10. Related to comment on section 3. Please clarify. At P4 L28-30, the 33 events are part of the 385 van Herwijnen and Schweizer (2011a) events. (sorry to insist ;))

---

## Author Comment (AC1) · 27 Sep 2017

The authors thank Referee #1 for the thorough and thoughtful review and the valuable comments. Please find our detailed reply in the supplement.

Please also note the supplement to this comment:
https://www.nat-hazards-earth-syst-sci-discuss.net/nhess-2017-224/nhess-2017-224-AC1-supplement.pdf

---

## Author Response (AR1)

**Repsonse to reviewer #1**

This study focuses on the seismic detection and identification of the signals generated by snow avalanches in the Davos area in the Swiss Alps during the winter 2010. The authors tested the capability of a machine learning algorithm (hidden Markov models - HMM) to perform this detection and identification from continuous seismic data. They used a reference catalog to evaluate the performance of the algorithm. The first results showed that the algorithm is capable to achieve relatively high positive identification rates of the avalanches in the catalog (70-95% depending on the station that recorded the signals), but also with a high rate of supposedly false detections. This led the authors to propose a post-processing strategy. Three post-processing steps were investigated: (i) analysis of the duration of the signals; (ii) computation of a correlation factor to evaluate the coherence of the signal between each sensor; and (iii) a voting system based on the classification returned by each station for a given event. Using one, or a combination of those proposed post-processing steps, led to a decrease of false alarm rates, but also in most cases to a decrease of the rate of good identification.

The use of seismology to study environmental processes is of growing interest as it allows producing observations with a unique spatio-temporal resolution. This new approach can help to better understand the triggering factors of natural hazards and to mitigate their consequences on our societies. In this context, this study contributes to the continuing effort to develop robust and versatile methods to explore years of continuous data and for the implementation of real-time seismology-based warning systems. Overall, I think the paper is clearly written, and that the Authors have made a good effort to carefully explore the data, explain their approach and discuss their results. Nevertheless I listed below several comments and suggestions that might help to improve this paper.

General comment:

The only major concern I have regarding this work is that the downsides of using the HMM algorithm are not discussed while most of the results presented in this paper suggests that HMM alone, without pre- and post-processing, cannot perform identification of seismic sources with a high success rate. The strengths of the HMM are usually stated to be:
- i) it does not need any pre-detection or picking (STA/LTA, etc.), which should ensure that no event is missed;
- ii) it does not require any inputs from experts.
Yet this paper demonstrates that
- i) a pre-detection can be suitable to remove low-amplitude/noise signals (figure 2);
- ii) post-processing steps with thresholds set by experts (duration, etc.) is needed to achieve a high accuracy.

Moreover, the Authors are building their post-processing strategy based on features that can be incorporated in the identification models constructed with other machine learning algorithms. The post-processing steps the Authors propose seem necessary because the HMM cannot include these features (durations of the signals, coherence between signals recorded at different stations, vote among stations) in the model due to its core design, which is to consider chunk of continuous data and not the entire signal generated by the event. This forces the Authors to manually set thresholds on those features, while with some other algorithms those thresholds are determined through a statistical analysis of the reference data. I think the authors must include a more thorough and objective discussion on the pros (which definitely exist) and the cons of HMM compared to other algorithms/studies in the light of the results of this work.

*We agree with the reviewer that we did not sufficiently discuss the shortcomings of HMM models. We now address this in more detail in the Discussion.*
*We would like to clarify that the signal amplitude threshold value we used should not be considered a pre-detection. We merely applied this threshold to reduce the data volume for the feature computation. Since the goal is to develop this method for operational application, reducing the computational time is of crucial importance. We would also like to point out that the HMM identifies events and returns a duration for the events. Event duration is thus a feature obtained by the model. Furthermore, the coherence between the sensors could also be used as a feature in the HMM. However, due to the high computational time required to calculate the coherence between 21 receiver pairs, we decided to use this feature only during post-processing. Calculating the coherence for a small number of detections is faster than calculating the coherence for the continuous data set. Finally, the reviewer is correct to state that the voting based classification cannot be included in the HMM.*

Specific comments:
P3 L6-8: Are those false alarm rates related to the choice of the algorithms or to the choice of the features used to parametrized the signals? The latter might be more important and should be mentioned.

*These false alarms rates are related to the choice of the algorithm.*

Figure 1: I think a colorscale with more colors would allow to better observe the features of the signals generated by the different sources, especially at frequencies below 50 Hz. This is important as the readers might want to understand what guided your choice of features. Also this figure can be larger.

*We chose this colour scale since it is perceptually uniform. We adapted the colour scale and enlarged the figure to more clearly highlight the features of the signals. However, the main goal of this figure was to highlight the ubiquitous environmental noise and not necessarily those of avalanche signals. The features of a typical avalanche signal are more clearly visible in Figure 7.*

P8 L25-27: How do you compute the duration?

*The duration of the avalanches in the reference catalogue was determined by visual inspection of the seismic data.*
*The onset was defined as the first appearance of energetic low frequency signals (i.e. between 15 and 25 Hz), while the end of the signal was defined as the time when low frequency signals reverted back to background levels (P5 L4-6).*
*The duration of any automatically detected event was determined by the HMM (P10 L9).*
*The minimal duration $T_{min}$ was determined manually and is described in the results section (P14 L12 – P16 L8).*

P9 L1-3: How does the voting step in the post-processing would impact the detection of "small" events (especially with a threshold set at 5 stations for a network with 7 sensors)? Are "small" events detected by the whole network? A figure showing the locations of the avalanche corridors and the seismic network would be interesting.

*We have added more details on the spacing between the geophones (see P 4 L5-9 ). A figure of the array can be found in van Herwijnen and Schweizer (2011a) and we refer the reader specifically to their figures (see Figure 3 and 4 in van Herwijnen and Schweizer, 2011a and Figure 2 in van Herwijnen and Schweizer, 2011b).*
*Due to the short distance between the sensors, the voting step does not neglect small avalanches. Signals of small events are recorded at all stations. The deployment of the sensors resulting in a less desirable SNR impacts more on the detection of small avalanches. Furthermore, most of the confirmed events can be regarded as small avalanches regarding international avalanche classifications.*

P11 L4-5: Indeed. How would it have impacted your results if you had chosen another master event? Is this has been investigated in Hammer et al. (2017)? If yes it should be mentioned and referenced.

*We used an avalanche event from the 100% group as master event. We compared it with other avalanche events and these all had similar features (Fig 5). We therefore can expect, that the classification results are similar.*
*We investigated the effect of using another training event and the overall results were very similar. However, since this is a very time consuming endeavour, we did not perform a more in-depth analysis of the influence of the training event.*
*Hammer et al (2012) investigated the dependence of classification performance on the reference event in detail. They show that the proposed approach is very robust in face of various master events.*

P13 L11-12: So the selection of the threshold on the duration is not done by considering the physics of the sources or the distribution of the durations in the reference catalog, but to optimize the POD-FAR ratio? By applying this threshold to the reference data set you lose 40% of the events (P14, last line). How is this threshold choice impacting the detection of "small" avalanches? Can you show a histogram of the duration of the events in the reference catalog? You state on P3 L24 that "For avalanche forecasting information on smaller avalanches is also required". Hence is the approach you propose suitable to detect the smaller events?

*Indeed, we optimized the duration threshold with the POD-FAR ratio. Since signal duration likely relates to avalanche size (van Herwijnen et al., 2013), it is clear that by increasing the signal duration threshold the detection rate of the smallest avalanches decreases. However, overall most of the avalanches in the reference catalogue can be considered to be 'small' for the purpose of avalanche forecasting. As such, we believe our approach is still suitable to detect smaller events.*

P13 L14-16, P20 L6-7, Table 3: Again, it would be great to have a map of the seismic network to discuss the discrepancies observed at the different sensors. Distance to the sources, traveled paths, attenuation, dispersion, etc., can also be factors impacting the amplitude, the duration and more generally the features of the signals that might in return change the POD at different stations. This could be discussed.

*Since the distance between the sensors was rather small (<12 m for the longest distance), we do not believe that local site effects substantially influence the signals. However, since the sensors were inserted in the snow, we are convinced that the main reason for the discrepancies in model performance between the different sensors is due to differences in snow cover properties, as discussed in P22 L5-10. To clarify this point in more detail, we now also mention in Section 2 that sensors 1 and 4 were most deeply covered by snow (P4 L5-8).*

P20 L17-18: So in the best case what is your overall accuracy? Considering which range of avalanche sizes? I think it is this information that the readers will seek.

*Overall accuracy is difficult to estimate. Although we have a reference catalogue, the actual number of avalanches in the continuous seismic data remains unknown. This is why we implemented the subjective probability classes in the reference data set. While such an approach does not provide a single performance value for the HMM, it shows that for clear avalanche signals (100% class) model performance is reasonable. Furthermore, it highlights the difficulties in obtaining a reliable and independent avalanche catalogue. However, in response to reviewer 2, we now also added information on the detection rate of the avalanches that were confirmed by images from the automatic cameras. These results show that at least 80% of the confirmed events were detected by all sensors.*

P20 28-30: Are the results presented in this study supporting this statement or is it based solely on the study by Hammer et al. (2017)?

*Our results support this statement since we also only used one training event for the HMM.*

P21 L4-5: How can you incorporate localization parameters in the HMM? Is this done directly in the model or during the post-processing?

*Both methods are possible. Localization metrics, in particular back azimuth and apparent velocity, of signals above the amplitude threshold could be used as feature for the HMM. On the other hand, it would also be possible to use the localization metrics during post processing by rejecting all detections having clear path.*

**Repsonse to reviewer #2**

General comments:

The paper discusses the performance of an automatic seismic detection system of snow avalanches using hidden Markov models (HMMs). The study is based on a 107- day continuous dataset acquired during the winter 2010 by a small seismic array consisting of 7 vertical geophones deployed above Davos, Switzerland, and surrounded by several avalanches starting zones. The HMMs system is tested against a reference avalanche catalogue that is based on the work by van Herwijnen and Schweizer (2011a). The reference avalanche catalogue gathers 283 events distributed into 7 probability classes following an independent re-evaluation of the avalanche signals of van Herwijnen and Schweizer (2011a) by three of the authors. Among the 283 events, only 20 (7%) are finally classified by the aforementioned three authors as being with 100% probability a snow avalanche. For the HMMs detector, the authors apply the approach developed by Hammer et al. (2012) that requires a background noise model and only one training event to learn the system. Because important diurnal and seasonal variations are observed in the seismic feature of the background noise over the season, a background noise model is recalculated for each day. Since the authors observe that the change from dry to wet avalanches through the winter does not have a strong impact on the selected seismic features used for the HMMs, only one avalanche signal is selected as a training event for a single avalanche class HMMs detector. The first HMMs avalanche detector results in high probability of detection (POD 70-95%) in the 100% probability class of the reference avalanche catalogue for all 7 stations. However, a large number (up to 2091) of additional events (unassigned detections) are detected by the HMMs for each sensors. In order to reduce the number of unassigned detections, the authors decide to introduce post-processing steps to the HMMs: (i) a duration based classification and (ii) an array based (multiple-station detection/coherence of signal among station pairs) classification. The postprocessing steps reduce the number of unassigned detections by the HMMs; however, the POD in the highest probability class of the reference avalanche catalogue is also reduced. A manual re-evaluation of the unassigned detections after post-processing shows that part of these detections correspond to avalanche missed during the manual detection procedure.

The paper is well structured and clearly written. Despite the noisy character of the data, the authors show HMMs as developed by Hammer et al. (2012) to be a promising approach for operational avalanche forecasting; however, with fine tuning. Strategies will be needed for a regular update of the background noise model and post-processing must be implemented. Much attention should be paid to the stations deployment. The author recommend to install the sensors 30-50 cm in the ground below a homogenous snow cover in order to reduce environmental noise and increase coherence of the data among the network and thus, stabilize the array based post-processing. In addition, a larger sensor inter-spacing would allow the system to apply array-processing techniques and enable to incorporate localization
parameters of avalanche source area in the automatic system. In my opinion, the implementation of the post-processing step is of great interest, as it helps to deal with the noise contamination that is found at any surface site data. The main concern I have is that the authors didn't make use of the 33 confirmed avalanches of van Herwijnen and Schweizer (2011a) and used instead a re-processed

probabilistic reference catalogue where only 20 avalanches remain considered as certain. In the following, a number of specific comments are related to this concern. I leave it to the authors to decide to which point they want to integrate these suggestions. It would be interesting to know how the 33 confirmed events match the post-processing steps: do they have durations above 12 seconds, are they detected by 5 sensors or more, what are the inter-station correlation indices for these events. In addition, if the location of the 33 confirmed events is known, how important are the variations in the spectral feature of the signals as a function of station-source distance? Finally, what is the performance of the HMMs system including the three post-processing steps against this original 385 events/33 confirmed events reference catalogue?

*We agree with the reviewer that adding the model performance for the confirmed avalanches would be insightful. We therefore updated figures (Figures 3 and 9) and tables (Table 1,2 and 3) to include these results and discuss them throughout the paper (e.g. P10 L10-11). Note that the number of confirmed avalanches originally mentioned in the manuscript was wrong and should have been 25. We believe that these results better highlight the model performance and we thank the reviewer for this valuable suggestion.*

Specific comments:

P1 L9-12. Please rephrase and clarify this part of the abstract (according to remarks about P4-7 Section 3).

*We rewrote this sentence (P1 L10).*

P4 L5-7. Although the station inter-spacing is specified in van Herwijnen and Schweizer (2011b), you should mention it here, so the reader has an idea of the size of the network.

*We now added this information in the text (P4 L6).*

P4-7. I think the section 3 needs some clarification so the reader can get a better idea of the reference data:

1) What are the exact data that were analyzed visually by van Herwijnen and Schweizer (2011a) (385 detection, 33 confirmed events): (1) 107-day of continuous data without pre-processing of section 3.1 at sensor number X or (2) XX% of pre-processed data (section 3.1) at sensor number X? In case (1), I would place P4 L28-30 before section 3.1 for clarity.

*We now clarify these points (P5 L1) but prefer to keep the sentence where it is.*

2) Section 3.1: The pre-processing of the data is obscure to me. Was it apply only to reduce the amount of data to (re-)process manually? Did van Herwijnen and Schweizer (2011a)/Heck et al. AND the HMMs perform only on the 20% of high

amplitude data (which would impact on the paper title as it would be no more continuous data)? If a local network is intentionally deployed to detect small avalanches, why place an energy threshold in the investigated data series? See for example avalanche number 5 (Av 5) in Figure 7 of van Herwijnen and Schweizer (2011b) which has poor SNR. Furthermore, HMMs are not dependent on energy thresholds, please clarify.

*The manual detection of avalanches was performed on the complete data set. However, calculating the features for all seven sensors and the entire data set is extremely time consuming. We therefore applied an amplitude threshold to reduce the amount of data by only taking those parts of the time series when some energy is arriving at the sensors. The amplitude threshold we used was very conservative and all the confirmed avalanches were still in the pre-processed data (Table 1 or Figure 3). Since the goal is to develop this method for operational application, reducing the computational cost is of crucial importance. As such, we do not believe that the pre-processing step we applied warrants a change of the title.*

3) Section 3.2: 20 merged avalanches represent 7% of the 283 reference events. In these data, where are the 33 confirmed avalanches of P4 L30? You could have used 33 confirmed events (9%) against 352 uncertain events of the original 385 events of van Herwijnen and Schweizer?

*We now include these confirmed events in Figure 3 and Table 1,2 and 3. Most of these events belong to the 100% or 83% class of the re-evaluated reference data set. For sensor 1, 20 of these events were detected and 5 missed.*
*For the voting based detection without using a minimal duration, also 20 were detected and 5 missed. By applying a minimal duration, the number of detections reduced to 18 hits and 7 missed.*

4) P6 L11-15: I think it would be interesting to discuss here in a few words the 283 events reference avalanche catalogue in more details. For example, do you find trends in the probability classes, are the 20 certain events confirmed detections, long duration signals, high energy events detected by all stations? Are the lower confidence event simply lower quality events (low SNR, only recorded at a few stations and not consistent among the stations)? These low probability events represent in my opinion the real challenge of environmental seismology case studies. And I expect the subjectivity of the analyst to have more influence on low quality data (how would/did you rate avalanche Av 5 in Figure 7 of van Herwijnen and Schweizer (2011b)) than on high SNR data? This is where automatic, quantitative systems should help :)

*We clarified these points by adding model performance statistics for the confirmed avalanche events throughout the text (Figures 9 and Tables 2 and 3). Furthermore, we added a figure (Figure 4 in the new manuscript) which shows the distribution of signal duration for each probability class. This figure clearly shows that the number of short duration events decreases with increasing probability class.*

5) P7 Figure 3. How does the original catalogue with 385 events, 33 confirmed avalanches plot? Are the two periods of higher activity well represented? I personally would insert a plot of the 33 confirmed events at the bottom of Figure 3.

*We now show the confirmed avalanches in Figure 4. Indeed, most of these avalanches also released during the high activity periods in March and April.*

P10 L8-20. Does this very sophisticated approach to find a threshold value bring more than an approach using common sense (for which arguments are well described in P8 L24-29 – P9 L1-20)? The minimum duration can be evaluated (or decided) as a function of the expected distance/size of the target events; the number of stations that must detect the signal can be selected as a reasonable value knowing the network/stations specific performances; the inter-station correlation threshold can be evaluated by investigating the 33 confirmed events. Please comment.

*We used a systematic approach to find threshold values by optimizing model performance. Since we had a reference avalanche catalogue this was possible. However, in the absence of such a reference data set, for instance when installing a system at a new location, we agree that to some extent it should be possible to select these thresholds based on some a priori available knowledge on the local topography and the array geometry. However, this requires some assumptions on the minimum avalanche size that can be detected as well as sensor performance.*

P11 L11-13. I agree that in Figure 5a the feature distribution is very similar for the 4 selected events. However, in Figure 5b, I find events of 21 Jan/21Feb dissimilar to events of 22Mar/24Apr, especially at the events onset where one group goes up while the other one goes down. Please comment on that and see remark on P15 Figure 8.

*Due to the different length of the signals, we decided to use a normalized time. As the events in March and April were quite long, the decrease for cepstral coefficient is longer. In January and March the decrease is shorter, thats why it seems as it only increases. This is also visible for the central frequency, where the frequency decreases slower for the March and April events.*
*Furthermore, for the HMM all 4 events are similar, since the feature behaviour (the decrease) is important. The absolute appearance of the features does not matter. This explains why the detection of events with different durations is possible.*

P12 Figure 5. In (a) and (b) vertical dotted lines at normalized time 0 and 1 would help to visualize start and end time of event.

*Figure 5 changed as suggested.*

P15 Figure 8. For both sensor 1 and 7, the HMMs missed 4 events at the end of January:
1) I don't find this 4-event spike in Figure 3?
2) Are any of the 33 confirmed avalanches found at this date? I see there is one 100% probability class event around this date in Figure 3.
3) Could this speak for an influence of the HMMs system by the avalanche type (dry-
wet) selected to train the system?

*The small spike in avalanche activity at the end of January was also present in Figure 3, marked as 33% avalanches. However, the spike is more obvious in Figure 8 since there we also applied the signal duration threshold to the reference data set. None of the confirmed avalanche events occurred on this day, as can now be seen in Figure 3 where we added the confirmed events.*

P18 L1-6. Clarification for the single sensor results:
1) Did you investigate the signals of the unassigned detections individually at station 1 and 6? Or, 2) Did you look at the signals recorded by all stations at the time of the unassigned detections of sensor 1 and 6?
I think these results should be commented. What could explain the higher false alarm rate of the array based approach against the single sensor approach?

*We investigated the results for each sensor individually. We clarified the sentence (P20 L1-2).*
*The higher false alarm rate is due to the higher signal-to-noise ratio for the other 5 sensors, as stated in the Discussion section (P22 L5-10).*

P19 L8-10. Related to comment on section 3. Please clarify. At P4 L28-30, the 33 events are part of the 385 van Herwijnen and Schweizer (2011a) events. (sorry to insist ;))

*The 25 confirmed events were part of the 385 original events and also remained in the 283 events 
[revised manuscript text omitted]